# How Much Agroforestry Is Needed to Achieve Multifunctional Landscapes at the Forest Frontier?—Coupling Expert Opinion with Robust Goal Programming

Esther Reith [1,*], Elizabeth Gosling [1], Thomas Knoke [1] and Carola Paul [2,3]

1   Institute of Forest Management, TUM School of Life Sciences Weihenstephan, Technische Universität München, Hans-Carl-von-Carlowitz-Platz 2, 85354 Freising, Germany; elizabeth.gosling@tum.de (E.G.); knoke@tum.de (T.K.)
2   Department of Forest Economics and Sustainable Land Use Planning, University of Göttingen, Büsgenweg 1, 37077 Göttingen, Germany; carola.paul@uni-goettingen.de
3   Centre of Biodiversity and Sustainable Land Use, Büsgenweg 1, 37077 Göttingen, Germany
*   Correspondence: esther.reith@tum.de; Tel.: +49-08161-71-5397

**Abstract:** Agroforestry has been promoted as a key forest landscape restoration (FLR) option to restore ecosystem services in degraded tropical landscapes. We investigated the share and type of agroforestry selected in an optimized landscape, accounting for a mosaic of alternative forest landscape restoration options (reforestation and natural succession) and forest and common agricultural land-uses. We extend previous studies on multi-objective robust optimization and the analytic hierarchy process by a systematic sensitivity analysis to assess the influence of incorporating agroforestry into a landscape. This approach accounts for multiple objectives concurrently, yet data and computational requirements are relatively low. Our results show that experts from different backgrounds perceive agroforestry (i.e., alley cropping and silvopasture) very positively. Inclusion of large shares of agroforestry (41% share of landscape) in the FLR mix enhanced simulated ecosystem service provision. Our results demonstrate that landscapes with high shares of agroforestry may also comprise of high shares of natural forest. However, landscapes dominated by single agroforestry systems showed lower landscape multifunctionality than heterogeneous landscapes. In the ongoing effort to create sustainable landscapes, our approach contributes to an understanding of interrelations between land-covers and uncertain provisions of ecosystem services in circumstances with scarce data.

**Keywords:** agroforestry; analytic hierarchy process; ecosystem services; forest landscape restoration; multifunctionality; optimization; uncertainty

## 1. Introduction

Agroforestry, the combination of trees and pasture or trees and crops on the same piece of land, is a promising system to reconcile ecological and socio-economic objectives in tropical regions [1–4]. For farmers and society as a whole, agroforestry may offer several advantages over conventional agriculture [4,5]. As a land-sharing strategy, agroforestry may be especially suited for re-integrating trees into degraded landscapes and has been discussed as a first step towards an agro-succession to increase forest cover [6,7]. Agroforestry systems, together with assisted natural reforestation and afforestation are among the forest landscape restoration (FLR) approaches [8]. FLR represents a landscape management strategy which aims to reconcile ecological and socio-economic objectives by restoring degraded agricultural and deforested lands [8–12]. By creating landscapes made up of

diverse and complementary land-use types, the objective of FLR is to restore ecological integrity and benefit human well-being [6].

In Panama, like in other tropical countries, old growth forest cover has been decreasing due to agricultural expansion [13–15]. Land abandonment in some parts of Panama led to slight net increase of forest cover due to natural secondary forest succession between 1992 and 2000 [15]. In an effort to reforest degraded land across the country, the Panamanian government has committed to one of the largest global restoration initiatives, the "Bonn Challenge", and partnered with private institutions in the national initiative "Alianza por el Millón de Hectareas Reforestadas" (Alliance for One Million Hectares Reforested) [9]. Existing restoration efforts in Panama have predominately focused on afforestation. For example, financial incentives for afforestation, enacted in 1992, have promoted commercial monocultures of teak (*Tectona grandis*), a fast-growing exotic species often owned by international timber corporations [16]. However, the expansion of these plantations has been criticized to mainly serve the objectives of large, mostly foreign reforestation companies, while rural needs, such as the need for frequent and regular cash flows, may be in conflict with this restoration option [16,17]. Furthermore, forest-plantations are sometimes called "green deserts", which reflects the debate around the biodiversity value of forest-plantations [18]. Therefore, in conjunction with Panama's reforestation project "Alianza por el Millón", there may be a shift toward more diversified reforestation options. For example, a new law specifically promotes agroforestry through tax exemptions and subsidies [19].

However, agroforestry may drive further deforestation if these systems prove economically competitive with profitable cropping or pasture systems [20]. Hence, decision-makers, such as landowners and landscape planning authorities, face the question of how much and which type of forest restoration option(s) is needed in different pre-existing landscape compositions or contexts to benefit landowners and the broader community [11]. This is a challenging task, given that ideally all land-uses of a landscape mosaic should be considered simultaneously to create a multifunctional landscape that fulfills multiple ecological and economic objectives, and avoids adverse consequences, such as deforestation.

Most research into multifunctional landscapes is positive in nature, aiming to describe and predict interactions between landscapes and ecosystem services. To illustrate, the impact of landscape structure on ecosystem services has been investigated through empiric statistical models (e.g., [21–23]) and system dynamics modelling (e.g., [24]). Agent based modelling has also been used to model decision-making of agents (e.g., farmers) and analyze interrelations of ecosystem services and land-use at the landscape scale (e.g., [25,26]).

While these approaches provide valuable information for landscape planning, our focus was in examining what a future landscape composition should look like to fulfill the objectives of multiple stakeholders. This concerns the uncertain provision of multiple ecosystem services (normative approach).

As a normative decision-support tool, multi-criteria optimization can be used to explore optimal land-cover compositions for reconciling multiple, potentially conflicting objectives [27,28]. In the case of our approach, this concerns ecological and socio-economic ecosystem service indicators. A common normative decision-support method is mean-variance optimization, based on modern portfolio theory. Portfolio theory is borrowed from financial sciences and builds on the premise that investing into different (not perfectly correlated) assets will reduce the overall portfolio risk. Translated to problems of land allocation, the method has been used to demonstrate the importance of high compositional diversity to stabilize economic returns but also to provide multiple ecosystem services [29–31]. However, these methods can be very data intensive due to the need to consider covariances among the criteria considered [32]. Portfolio theory in the context of land allocation has furthermore mainly been applied to optimize a single, usually economic objective, but has rarely been coupled with multiple objective functions [30,33]. As an alternative to mean-variance optimization, robust portfolio optimization does not require specific knowledge on correlations. Furthermore, robust optimization is less data-demanding when accounting for perturbations or uncertainty, which stem from the underlying variation in the provision of ecosystem services in our context [20]. While stochastic

mean-variance optimization assumes probabilistic uncertainty, robust optimization is deterministic. Considering numerous constraints to account for all input data included in so-called uncertainty sets, robust optimization finds a solution which guarantees that none of the constraints are violated [20,34]. Knoke et al. [35] developed a robust optimization model to optimize land-cover diversification that provides multiple ecosystem services, while reducing trade-offs between them. They investigated land allocation to provide socio-economic benefits and ecological functions in Ecuador [35].

To represent multiple objectives in land-use modelling, indicators can be used. Indicators help to assess changes in ecosystem services owing to changes in land-use practices [36]. For example, the status of biodiversity has been assessed using an indicator that uses land-use composition as proxy for potential habitats within a given landscape and relates this to the level of biodiversity within that area [37,38]. Datasets from field trials, remote sensing data and from approved databanks, such as those available for InVest [39,40], are valuable tools for quantifying many ecosystem services. Other important socio-economic objectives, such as expected profits, economic stability or cultural preferences, may be difficult to assess without involving stakeholder groups. In particular, perception of landscape value is not easily quantifiable, but may be important to include [41]. In addition, comprehensive datasets for ecological and socio-economic indicators for many land-cover types, including FLR, are seldom available.

As an alternative to measured field data, expert knowledge has been applied to estimate the performance of land-cover types in terms of ecological and socio-economic services [42–45]. For example, Lima et al. [46] combined remote sensing with expert knowledge to map ecosystem services in the Brazilian Savanna and to assess the impact of landscape properties on providing ecosystem services. While their approach has the advantage of not relying on complex modelling tools, it cannot inform about desirable future landscape compositions, including information on land-uses currently not practiced.

Fontana et al. [47] evaluated ecosystem service provisions across three land-use alternatives in the central European Alps, eliciting stakeholder opinion via the analytic hierarchy process (AHP; Saaty [48]). AHP is a popular multiple criteria decision support method that allows expert knowledge to be transferred to a ratio scale [43]. Through pairwise comparison, experts estimate the relative importance of items [48]. For example, Uhde et al. [49] asked experts to compare five forest management options in Chile in terms of ecosystem service provision using AHP. They used the quantified expert knowledge as input data for multi-objective robust optimization based on the model by Knoke et al. [35].

Our study deals with the important challenge of allocating land to different land-cover types while considering trade-offs between them. We intend to better understand the interrelations between different land-cover alternatives and landscape compositions for providing ecosystem services. The optimized landscape compositions might provide a useful starting point for landscape planning and stakeholder discussions, to agree on what an optimal landscape might look like, and to see how these optimal landscapes may change under different pre-existing land-use mosaics. This study advances on previous studies in determining how much of single restoration options is judicious to meet ecological objectives, while being socio-economically attractive and robust in the face of future uncertainties. We couple expert-interviews using AHP with multi-objective robust optimization, but extend the Uhde et al. [49] study, which is limited to forestry, to a landscape approach by considering natural forest, agricultural land-uses and different FLR options including agroforestry. The main contribution of our study is an extensive sensitivity analysis to investigate the potential of agroforestry and other FLR options to increase ecosystem services under various landscape compositions. Previous incentives led to an expansion of forest-plantations, making it the most widespread FLR option in eastern Panama [16]. Therefore, we were interested in analyzing the effect of increasing shares of single land-cover types on optimal land allocation of the remaining landscape and its multifunctionality. This includes exploring the impact of promoting agroforestry on the composition of the remaining landscape and on ecosystem service provision of the entire landscape.

Thus, this study is guided by three research questions:

1.  How much agroforestry would be desirable in a mix of FLR options to balance ecological and socio-economic ecosystem services at the landscape scale under uncertainty?
2.  How does the landscape context impact the share of agroforestry under uncertainty?
3.  How does the promotion of agroforestry affect the remaining landscape composition under uncertainty?

## 2. Materials and Methods

### 2.1. Study Area

We exemplify our approach with a study area at the forest frontier of eastern Panama. Our study area covers around 9100 ha, centering of the rural township of Tortí, which belongs to the Chepo District and is located on the Pan-American Highway about 25 km from the border between the Panama and Darien provinces. Fifty years ago, this region was covered by rainforest [50]. Nowadays, the landscape consists of pasture (46%), exotic forest-plantation (22%), cropland (20%) and only a small remnant of natural forest (12%) (see Supplementary Method S1).

### 2.2. Estimating Ecosystem Services Provided by Land-Cover Types

To capture the performance of a landscape for meeting multiple objectives, we used 10 ecosystem service indicators to evaluate ecological and socio-economic objectives (Table 1). To identify relevant ecosystem service indicators, we conducted a literature search and validated the final set of indicators with experts in the pre-test of our survey. The ecological indicators reflect the capacity of a given land-cover for hydrological and climatic regulation, supporting biodiversity and soil fertility. The socio-economic indicators address direct benefits to humans. Among them are the stable provision of food (food security), financial performance (long-term profit, liquidity and stability of economic return) and an aesthetic landscape for society. Our selected indicators cover the four classes of ecosystem services defined by the Millennium Ecosystem Assessment [51]: regulating, supporting, provisioning and cultural. Acknowledging that biodiversity is not an ecosystem service in a strict sense [52], we refer to biodiversity conservation as an additional objective associated with habitat provision. We recognize that there is uncertainty around which ecosystem services will be demanded in the future, and therefore examined a large set of indicators [28].

**Table 1.** Description of the ecosystem service indicators. They represent the objectives in robust multi-objective optimization.

| Category | Ecosystem Service Indicators | Description |
|---|---|---|
| Ecological | Global climate regulation | Contribution of land-cover to regulate global climate, i.e., the capacity of vegetation to store atmospheric carbon (without taking into account substitution effects). |
| | Water regulation | Contribution of land-cover to regulate water flow and supply, e.g., reduced overland flow. |
| | Biodiversity | The extent to which the land-cover supports species richness, i.e., the number of plant and animal species. |
| | Long-term soil fertility | Capacity of land-cover to maintain soil fertility, protect soil quality and soil health over the long-term (e.g., 20 years). Potentially quantified through carbon-nitrogen-ratio. |
| | Micro climate regulation | Contribution of land-cover to local and regional climate regulation. For example, the effect of trees on air temperature and wind speed [53]. |

**Table 1.** *Cont.*

| Category | Ecosystem Service Indicators | Description |
|---|---|---|
| Socio-economic | Food security | The extent to which the land-cover type provides a stable food supply concerning dietary calories produced. |
| | Long-term profit | Contribution of land-cover to provide income in the long run (e.g., 20 years). Potentially quantified through the present value of cash flows generated by the land-cover over time. |
| | Liquidity | The extent to which the land-cover provides frequent and regular income flows, including how easily the land-cover can be converted to cash if needed. |
| | Stability of economic return | Contribution of land-cover to provide stable returns against risk (e.g., extreme weather events, price fluctuations). Potentially quantified through financial losses. |
| | Scenic beauty | The extent to which the land-cover provides an aesthetic landscape for society. |

We analyze seven land-cover types in this study (Table 2). This includes the two purely agricultural land-cover types, cropland and pasture, as well as natural forest and four FLR options. In our study, FLR options entail afforestation and regeneration of deforested and degraded landscapes, as well as reintegrating trees in productive units through agroforestry [54]. Common FLR options in eastern Panama are commercial forest-plantation [31] and natural succession of abandoned land [55]. Potential new FLR options are alley cropping and silvopasture agroforestry systems, as defined in Table 2. We selected alley cropping because it can be expanded at different scales. Although not common in the study region, local trials coupled with bio-economic modelling found alley cropping to be an economically competitive land-cover type [31]. We focus on an alley cropping system with a tree and a crop component instead of considering that, with time, the tree canopy would close and annual crop production cease. This is because our analysis is static and does not consider time dynamics. Silvopastoral systems with living fences and scattered trees are common in the study region [56]; however, we were interested in a system with a higher tree density, which can be used for timber production. As stocking rates and tree densities per hectare vary in the literature [57–59], we opted for a conservative number of cattle and trees per hectare (Table 2).

**Table 2.** Description of the land-cover types. Superscript denotes the FLR options.

| Land-Cover | Description | Source |
|---|---|---|
| Cropland | Cropland can include various species of annual crops. Different crops might be cultivated at the same time on one plot of land (crop-mix) or rotated over a time (crop rotation). For planting and harvesting, farmers mainly use manual/traditional methods. | [56] |
| Pasture | Traditional pasture with a stocking rate of one and a half to two cows per hectare, can include scattered trees. | [50,55] |
| Alley cropping[FLR] | An agroforestry practice where alleys of trees (with a distance of around 6 m between trees) are alternated with rows of annual crops. Trees are grown for timber. | [31] |
| Silvopasture[FLR] | An agroforestry practice where cattle (conservative count of around one cow per ha) and trees (around 200 trees per ha) are combined on the same plot of land. Trees are planted or guarded against cows and harvested for timber. | [57,60] |
| Forest-Plantation[FLR] | Forest-plantations comprising one introduced tree species (e.g., teak, *Tectona grandis*) forming even-aged stands and planted with regular spacing ($3 \times 3$ m). Trees are pruned, thinned and harvested. | [31] |
| Abandoned land[FLR] | Natural succession of abandoned land: Agricultural land (cropland or pasture) which has not been managed or cultivated for more than five years, mainly due to low productivity. There can be secondary succession of vegetation. | [55] |
| Forest | Humid tropical forest, specifically unmanaged secondary forest with natural regeneration. Forest is neither under conservation (i.e., can be used to collect firewood or fruits for human consumption), nor managed for commercial purposes (i.e., timber production). | [50,55] |

To estimate the investigated ecosystem services provided by the selected land-cover types, we conducted expert surveys. To ensure that our sample represented an informed view, we used a stratified, purposive sampling approach [61] to target experts from five stakeholder groups: universities and research institutes, government agencies, non-government organizations (NGOs), corporations and farmers and local residents in the study region (who can also be considered shareholders). To identify relevant experts, we contacted organizations and institutes that contributed to a major environmental publication in Panama, the "Atlas Ambiental de la República de Panamá" (The Republic of Panama Environmental Atlas) [62]. We used these initial contacts to broaden our sampling frame through snowball sampling [61]. This allowed us to purposively select experts in pertinent organizations and institutions that hold a position relevant to our research, followed by the use of a primary sample to expand our research by including further relevant participants. We included experts who currently or have previously worked in Panama, and who had expertise in at least one of the following fields: agriculture, agroforestry, biodiversity, climate science, economics, forestry, hydrology and soil science. The field of expertise determined which ecosystem service indicator experts estimated (for further details see Method S2). International experts (with experience in Panama) were sourced by contacting authors of relevant literature. We targeted farmers and local residents by approaching randomly selected houses in the study area and asking the inhabitants if they manage a farm or have a background in farming. If they had that experience, they were asked if they would be willing to participate in the survey. A full breakdown of the number of respondents per indicator and stakeholder group is given in Table S2. We surveyed experts from April to September 2018.

During the survey, we used AHP to generate rankings of the land-cover performance against each of the ecosystem services as perceived by the experts. AHP decomposes complex decision-making processes into a series of pairwise comparisons. Survey participants were asked to complete 21 comparisons of seven land-cover types for each ecosystem service indicator. The output of the AHP survey were mean scores for each land-cover for each indicator. We aggregated the individual results across all respondents to obtain a group judgement reflected by the mean, and their standard deviation. Scores can range from 1 to 17, where high scores signify a land-cover which was better able to achieve a given ecosystem service indicator than the land-cover used for comparison (Table 3). The generated performance data of the land-cover types formed the input data for the optimization model (see below). An advantage of AHP is that it enabled us to consider a wide range of objectives, including those that are not easily quantifiable, such as scenic beauty. Details of the approach used can be found in the Supplementary (Method S2).

A total of 54 representatives from 36 organizations and 26 farmers and local residents participated in the survey. We obtained 36 to 40 evaluations per ecosystem service indicator, where an evaluation represents a completed set of pairwise comparisons (Table S2).

A strength of AHP is its ability to include various stakeholder groups and different techniques. We used two techniques to conduct the AHP survey: an online survey and face-to-face interviews. In both cases, we provided participants with information about the purpose of the research before starting the survey, and we informed them that their participation was voluntary and all answers confidential. The introduction of the survey also included information on the study region and the definitions of each indicator and land-cover (Tables 1 and 2).

A comparison of the mean indicator scores derived from the two survey methods showed no noteworthy differences between results (Figure S2). For instance, when comparing the aggregated mean scores of the online and face-to-face survey of the pairwise comparisons of two land-covers including 10 indicators, 69% of the comparisons had a difference of ±1 on a scale from 1–17 (Figure S2). Twenty-four percent of the aggregated mean scores of the online and face-to-face interviews had a difference greater than ±1, but lower than ±2.5. The remaining 7% of the mean scores differed by ±2.5 to ±5.5, which would not significantly impact the overall results.

**Table 3.** Ecosystem service indicator scores for land-cover types. Scores were derived from AHP survey and used as input data for multi-objective optimization to obtain a theoretically optimal landscape composition. Figures represent expected mean scores and standard deviation (in parentheses). N is the number of survey participants considered per ecosystem service indicator. The higher the mean score, the more important the land-cover for a given indicator (score range 1 to 17). Highest mean scores for each indicator are given in bold.

| Category | Ecosystem Service Indicators | Cropland | Pasture | Alley Cropping | Silvopasture | Forest | Forest-Plantation | Abandoned | N |
|---|---|---|---|---|---|---|---|---|---|
| Ecological | Global climate regulation | 5.2 (±1.42) | 4.2 (±1.48) | 10.1 (±1.70) | 9.0 (±2.10) | **15.4** (±1.46) | 12.1 (±2.31) | 7.0 (±2.92) | 40 |
| | Water regulation | 5.5 (±1.44) | 5.0 (±2.05) | 10.2 (±2.20) | 9.2 (±1.84) | **15.4** (±2.26) | 10.4 (±2.10) | 7.3 (±3.28) | 39 |
| | Biodiversity | 5.2 (±1.37) | 4.6 (±1.53) | 10.0 (±1.78) | 9.0 (±1.56) | **16.1** (±1.03) | 9.5 (±2.49) | 8.6 (±3.55) | 38 |
| | Long-term soil fertility | 5.6 (±1.23) | 4.8 (±1.81) | 9.9 (±1.84) | 8.7 (±1.74) | **15.8** (±1.89) | 9.6 (±2.59) | 8.5 (±3.31) | 38 |
| | Micro climate regulation | 5.1 (±1.17) | 4.7 (±1.41) | 10.4 (±2.00) | 9.0 (±1.74) | **15.7** (±1.25) | 10.9 (±1.95) | 7.1 (±3.29) | 38 |
| Socio-economic | Food security | 11.3 (±3.98) | 8.7 (±2.77) | **12.8** (±2.11) | 11.9 (±2.37) | 7.9 (±2.93) | 5.9 (±2.20) | 4.7 (±2.38) | 36 |
| | Long-term profit | 7.9 (±3.23) | 7.9 (±2.83) | **12.2** (±2.39) | 11.9 (±2.06) | 8.0 (±3.92) | 10.9 (±3.05) | 4.2 (±2.27) | 37 |
| | Liquidity | 11.5 (±2.98) | **11.6** (±2.35) | 10.8 (±2.64) | 11.2 (±2.46) | 6.4 (±3.55) | 7.3 (±2.67) | 4.2 (±2.79) | 37 |
| | Stability of economic return | 7.6 (±3.16) | 7.8 (±3.02) | 11.1 (±3.08) | **11.1** (±2.03) | 9.6 (±3.42) | 10.0 (±3.23) | 5.7 (±3.94) | 36 |
| | Scenic beauty | 6.7 (±2.24) | 6.4 (±2.49) | 12.1 (±2.53) | 11.3 (±2.16) | **12.5** (±3.23) | 9.8 (±2.61) | 4.3 (±2.25) | 37 |



Since we were analyzing a multifunctional landscape, we weighted all stakeholder groups and their rankings equally. We refrained from weighing experts to avoid bias, but to account for variability of expert answers, we included the standard deviation of indicator scores in our optimization. We explicitly investigate how the agreement and disagreement of experts about the relative provision of different ecological and socio-economic objectives affect the theoretical optimal landscape composition.

## 2.3. Optimization Approach

To find the optimal mix of land-cover types for securing a multifunctional landscape, we turn to robust multi-objective optimization. The input data for the optimization is the experts' evaluation of the ability of the seven land-cover types to provide the 10 ecosystem services (Table 3). The optimization model can simultaneously consider all studied land-cover types and potential fluctuations in their contribution to 10 ecosystem services, which cannot necessarily be predicted by experts during the survey.

Our optimization method is a variant of goal-programming implemented as a linear program to obtain an exact solution [63]. The goal-programming approach is coupled with a robust optimization to incorporate uncertainty in the decision process [64]. This normative approach suggests how land management can be improved to balance the achievement of multiple ecosystem services in eastern Panama. While optimization can be used positively to represent current land management or make predictions [65,66], our study is intended to illustrate how land-covers should be reallocated to better meet a pre-defined set of objectives (i.e., ecological and socio-economic ecosystem service indicators) and constraints, described below. Table 4 outlines the key variables of the optimization model.

**Table 4.** Overview and description of variables in multi-objective optimization model.

| Variable | Description |
| --- | --- |
| $i$ | ecosystem service indicator |
| $l$ | land-cover type |
| $R_{li}$ | nominal score of ecosystem service indicator, $i$, provided by land-cover, $l$, derived from the AHP survey |
| $SD_{li}$ | standard deviation of nominal score for ecosystem service indicator, $i$, and land-cover, $l$ |
| $fu$ | uncertainty factor to determine the deviation from the expected nominal score, $R_{li}$, ranging from 0 (ignoring uncertainty) to 3 (high level of uncertainty) |
| $u$ | uncertainty scenario |
| $R_{liu}$ | score of ecosystem service indicator, $i$, for land-cover, $l$, adjusted for uncertainty, $u$ |
| min $\{R_{liu}\}$ | minimum uncertainty-adjusted indicator score, $R_{liu}$, across all land-cover types in a given uncertainty scenario |
| max $\{R_{liu}\}$ | maximum uncertainty-adjusted indicator score, $R_{liu}$, across all land-cover types in a given uncertainty scenario |
| $R_{iu}$ | represents the sum of the ecosystem service indicator scores for each land-cover type, weighted by their area share in the landscape composition for each uncertainty scenario |
| $a_l$ | allocated share (area fraction) of a given land-cover type, $l$, in a landscape composition |
| $p_{iu}$ | normalized indicator score, $i$, for a landscape composition per uncertainty scenario, $u$, expressed as a percentage (landscape performance value)—100% represents best possible performance |
| $D_{iu}$ | distance between the normalized indicator score, $p_{iu}$, of a given ecosystem service indicator, $i$, and the hypothetical maximum of 100% (can be thought of as underperformance) |
| $\beta$ | maximum underperformance, $D_{iu}$, across all indicators, $i$, and all uncertainty scenarios, $u$ (worst underperformance) |

As a first step, we used AHP to derive nominal values ($R_{li}$) of ecosystem service provision for each land-cover type, $l$, and indicator, $i$. Together with their standard deviation ($SD_{li}$), these scores represent the input values for our optimization model (Table 3). Through the standard deviation, we incorporate potential deviations from the expected nominal indicator value and account for uncertainty in the ability of the studied land-cover types to achieve a given ecosystem service indicator.

Uncertainty reflects two phenomena in our study: a lower consensus among experts (standard deviation, Table 3) and a lower predictability of the provision of the ecosystem services by the respective land-covers (multiplication of standard deviation with uncertainty factor $fu$, Equation (1)). With our treatment of uncertainty, we address "deep" uncertainty in our modelling, which Walker et al. [67] denote as level 4 uncertainty. Beyond this level of uncertainty is total ignorance. Deep uncertainty means that we are neither able to specify probabilities nor to provide exact rankings regarding the performance of each land-cover type for achieving each indicator. Consequently, we address

uncertainty through uncertainty sets, defined by unique combinations of optimistic and pessimistic values for each indicator achieved by our land-cover types. In total, we incorporated 128 ($2^7$ for seven land-cover types) uncertainty scenarios ($u$) for each of the 10 ecosystem service indicators following Knoke et al. [28] (for an example see Supplementary Figure S3). We include the nominal (mean) score $R_{li}$ as our best case and compute an unfavorable deviation of this score as our worst case (Equation (1)).

$$R_{liu} = R_{li} \text{ for best case}$$
$$R_{liu} = R_{li} - fu \times SD_{li} \text{ for worst case} \tag{1}$$

This way, we only consider unfavorable deviations from the expected (nominal) value and minimize underperformance in worst-case scenarios. Unfavorable deviations are computed by subtracting multiples of $fu$ of the standard deviation from the mean score $R_{li}$ (Equation (1)). A value of 0 for $fu$ ignores uncertainty, whereas a value of $fu = 3$ represents a high level of uncertainty and risk aversion of a decision-maker in landscape planning. We ran the optimization for $fu = 0, 0.1, 0.2, \dots 3$. The ability of a given landscape composition to provide a given indicator under uncertainty is represented by the indicator level achieved for each uncertainty scenario ($R_{iu}$). We computed $R_{iu}$ by weighing the indicator values adjusted for uncertainty ($R_{liu}$) with the shares of the total land area allocated to each land-cover type within that composition ($a_l$) (Equation (2)), with the constraints $\Sigma a_l = 1$ and $a_l \geq 0$.

$$R_{iu} = \Sigma R_{liu} \times a_l \tag{2}$$

We then normalize all indicator scores achieved per uncertainty scenario ($p_{iu}$) between the minimum (min $\{R_{liu}\}$) and the maximum (max $\{R_{liu}\}$) indicator scores within each uncertainty scenario. The derived value is given as a percentage (Equation (3)).

$$p_{iu} = (R_{iu} - \min \{R_{liu}\})/(\max \{R_{liu}\} - \min \{R_{liu}\}) \times 100 \tag{3}$$

Finally, we calculate the distance ($D_{iu}$) between the indicator value (achieved and normalized, $p_{iu}$) and the hypothetical maximum of 100% (Equation (4)) for each uncertainty scenario and indicator, where $0 \leq D_{iu} \leq 100$:

$$D_{iu} = 100 - p_{iu} \tag{4}$$

where $p_{iu}$ is the normalized indicator performance value expressed as a percentage. The uncertainty scenario with the lowest performance value (highest $D_{iu}$) across all indicators then determines the maximum distance $\beta$ to the hypothetical maximum (100%). The model seeks to minimize this maximum deviation $\beta$ from the maximum achievement level among all indicators and uncertainty scenarios by allocating land to the different land-cover types. In other words, the optimization problem aims to minimize the worst underperformance:

Objective function:

$$\min \beta \tag{5}$$

with

$$\beta = \max \{D_{iu}\} \tag{6}$$

subject to:

$$\beta \geq D_{iu} \text{ (for all } i \text{ and } u) \tag{7}$$

The inequation (Equation (7)) summarizes individual constraints (here, 128 constraints: one for each uncertainty scenario, ×10 indicators), with $\beta$ (the objective function) as the maximum tolerated distance on the left side of the inequation, and $D_{iu}$ as the actual distance to the maximum achievement level on the right side. To solve the allocation problem, the land-cover weights $a_l$, the left side of the constraints (Equation (7)) and the objective function $\beta$ are defined as changeable variables. The problem can then be solved by the Simplex algorithm offering an exact solution for a compromise land-cover composition that minimizes the worst underperformance across all ecosystem service indicators.

Therefore, we used the Frontline Solver V2017-R2 (17.5.1.0) (Frontline Systems Inc., Incline Village, Nevada, USA) to run the optimization in a Microsoft Excel environment, but an open source software can also be used (e.g., OpenSolver (2.9.0) (Department of Engineering Science, University of Auckland, Auckland, New Zealand)).

When optimizing for a desirable landscape, we do not allow high performance in one objective (here indicator) to compensate for poor performance in another [28]. For example, high species richness of plants and animals (our biodiversity indicator) cannot compensate for low food security. Thus, the optimized landscape represents a compromise solution that meets all 10 objectives concurrently. This solution reflects the best performance for the worst-case uncertainty scenario across all ecosystem service indicators. Our optimized landscape portfolios represent a suggestion for a desirable future land allocation to best provide a compromise solution that meets the needs of a large group of stakeholders (normative perspective), rather than predicting what a future landscape composition would look like (positive perspective).

The optimization approach weighs all objectives equally. We abstained from weighing specific indicators to derive an objective compromise solution, which could then be used as a baseline for further participatory approaches.

### 2.4. Analysis of the Landscape Context

To better understand the interrelations between the landscape composition and agroforestry in terms of ecosystem services provision, we conducted a systematic sensitivity analysis. We aimed to provide insight on which mix of FLR options might be best-suited under different hypothetical land-cover contexts. To simulate different landscape contexts, we increased the shares of forest, forest-plantation, natural succession of abandoned land and agricultural land in steps 0, 0.1, 0.2, ... 1 imposed through a constraint, considering a moderate level of uncertainty ($fu = 2$). This allowed us to simulate landscapes covered with large shares of single land-cover types. We then examined the optimized composition of the remaining landscape portfolio (not occupied by the single land-cover type) and the ecological and socio-economic impact. Similarly, we increased the area share of the two agroforestry types (alley cropping and silvopasture) to understand the effect of promoting agroforestry as one FLR option.

We analyzed the overall landscape performance in terms of ecosystem service provision (min $\{p_{iu}\}$). We derived the guaranteed performance level achieved across all indicators and uncertainty scenarios by calculating the distance between the guaranteed level of the ecosystem service indicator to the hypothetical maximum indicator value:

$$\min \{p_{iu}\} = 100 - \beta \tag{8}$$

We also assessed the compositional landscape diversity of the optimized landscape portfolios. Using Shannon's index [68], we compared the diversity of the remaining land-covers in a portfolio, when one land-cover type dominated the landscape. We calculated the diversity index for each land-cover portfolio as follows:

$$H = -\Sigma \, a_l \ln a_l \tag{9}$$

where $a_l$: is share of land-cover, $l$, in a given landscape portfolio.

## 3. Results

### 3.1. Agroforestry and Other FLR Options to Balance Ecological and Socio-Economic Objectives

Based on expert opinion, under a moderate level of uncertainty, a large share of agroforestry (41%) was selected to complement other FLR options, natural forest and agricultural land-cover types to balance all studied ecosystem services simultaneously in a landscape mosaic.

Overall participants rated forest as the best land-cover type for achieving the ecological indicators, while agroforestry scored highly for the socio-economic indicators (Table 3). Pasture, which is currently the most common land-cover in the study area, was only selected as the best land-cover type for the socio-economic indicator liquidity (mean score 11.6 out of 17). The lower standard deviation of the ecological indicator scores suggest a higher level of consensus in expert opinion for this group of indicators (coefficient of variations ranged between 6% and 46% compared to 16% to 69% for socio-economic indicators (Table S3)).

Among the four FLR options, both agroforestry systems were perceived by experts as the best two land-cover types to provide food security, long-term profit and stable economic returns (Table 3). Generally, agroforestry was ranked best or second best for 6 out of 10 ecosystem service indicators investigated. Hence, agroforestry was consistently part of the optimized landscape for different levels of uncertainty in providing studied ecosystem services, from ignoring uncertainty ($fu = 0$, see Equation (1)) up to a high level of uncertainty ($fu = 3$) (Supplementary Figure S4). In contrast, natural succession of abandoned land was not part of the landscape portfolio, whereas forest-plantation was selected by the model only from a level of uncertainty of $fu \geq 1.5$. These two FLR options were not perceived as the best approaches for providing any ecosystem service indicator (Table 3).

Our results showed that the level of uncertainty affects landscape diversity. At lower levels of uncertainty, the landscape would comprise large shares of either alley cropping or silvopasture. For an uncertainty level of $fu \geq 1.5$, the landscape comprises increasingly equal shares of six land-cover types (Supplementary Figure S4). In the following sections, we focus our analysis on a moderate uncertainty level ($fu = 2$). This means that the considered deviation from the expected score of the ecosystem service indicator is twice as large as the standard deviation of the indicator.

We found that the theoretically ideal landscape composition under a moderate level of uncertainty diverges strongly from the actual land-cover composition in the study area (compare left and right-most columns in Figure 1). In the current landscape, forest and agricultural land-covers were complemented by one FLR option only: forest-plantation. Pasture represented the greatest area share (46%). However, in the optimized landscape, pasture and cropland only comprised a 12% and 9% share, respectively. The remaining area was assigned to forest and FLR options with a large share of agroforestry (41%). When agroforestry systems were excluded from the optimization model, the optimized land-cover composition became more similar to the actual composition, but cropland was substituted by abandoned land (Figure 1). This is likely because natural succession of abandoned land (which can be thought of as recovering secondary forest) was perceived to perform better in terms of ecological indicators compared to cropland and pasture.

Apart from studying the landscape performance in terms of balancing all 10 indicators for ecosystem services simultaneously, we examined the achieved performance level of the individual ecosystem services separately (Figure S5). For the optimal landscape portfolio including agroforestry (left column in Figure 1), the worst performing indicators were water regulation, food security, liquidity and economic stability (indicator values achieved $p_{iu} \geq 35\%$, Figure S5). This means that across all uncertainty scenarios, the 10 indicators achieved a performance level of at least 35% (where 100% is the hypothetical maximum). In comparison, excluding agroforestry resulted in a lower guaranteed ecosystem service indicator level (center column, Figure 1), with the poorest performance for food security, long-term profit and scenic beauty ($p_{iu} \geq 15\%$), closely followed by economic stability (Figure S5). This was due to the strong performance of agroforestry for those four ecosystem service indicators. For the current landscape portfolio, economic stability (closely followed by food security) was the worst performing indicator (with a guaranteed performance level of only 9% (Figure S5)).

In addition, we performed single-objective optimization, i.e., we determined the optimal land allocation for achieving each ecosystems service indicator individually instead of all indicators simultaneously (Supplementary Figure S6). As expected, landscape performance was higher when optimizing for single indicators. For example, a landscape entirely covered by forest may achieve the hypothetical maximum ecosystem service level (100%) for single ecological ecosystem service

indicators. Optimized landscapes for single socio-economic indicators achieved guaranteed ecosystem service levels between 45% and 61% and were dominated by agroforestry systems (Figure S6).

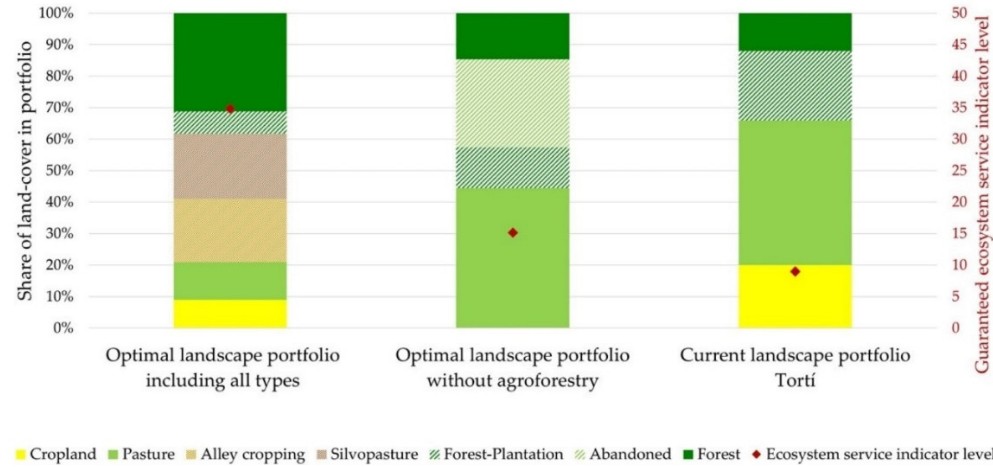

**Figure 1.** The composition and performance of optimized landscape portfolios (left and middle columns) and the current land-cover allocation in the study region (right column). Left axis shows the area shares of each of the seven land-covers. Right axis (red diamonds) shows the guaranteed level of ecosystem service indicators (min $\{p_{iu}\}$, see Equation (8)) for each portfolio. The optimized portfolios are derived for a moderate level of uncertainty ($fu = 2$), when including (left column) and excluding (middle column) agroforestry from the multi-objective optimization.

### 3.2. Influence of Landscape Context on Agroforestry Selection

The sensitivity analysis showed how much agroforestry would be desirable in varying landscape contexts to balance multiple objectives under uncertainty (Figure 2). We analyzed the share of agroforestry, landscape diversification (Figure 2, stacked columns, left *y*-axis) and the performance of optimized landscape portfolios regarding ecosystem service provision (Figure 2, red line, right *y*-axis) for landscapes dominated by either (a) cropland, (b) pasture, (c) natural forest or one of the FLR options, (d) forest-plantation or (e) natural succession of abandoned land (Figure 2, *x*-axis). The resulting landscape portfolios may be interpreted as the desirable land-cover allocation when following expert opinion, for situations in which single land-cover types are already widespread in the landscape.

To increase the level of ecosystem service indicators within the optimized portfolios, the model consistently selected a mix of FLR options including agroforestry when progressively increasing the share of a single (non-agroforestry) land-cover types. This suggests that a mix of FLR options, natural forest and agricultural land-uses are needed to balance the achievement of all 10 ecosystem services, irrespective of the dominant land-cover type (Figure 2). For example, to secure the highest guaranteed level of multiple ecosystem services an increase in forest-plantation share was not compensated for by an increase in cropland or pasture, but instead by allocating land to a mix of land-cover types including a large area of agroforestry (45% to 55% agroforestry share of the remaining landscape portfolio, Figure 2d).

Agroforestry comprised 50% to 66% of the remaining land-cover portfolios for landscapes dominated by cropland (Figure 2a) or with a forest share larger than 30% (Figure 2c), or forest-plantation share larger than 20% (Figure 2d). For increasing shares of pasture (Figure 2b) or abandoned land (Figure 2e), agroforestry shares comprised 34% to 49% of the remaining landscape.

When progressively increasing the share of single land-cover types, the share of agroforestry in the remaining portfolio was stable, except for the landscape with increasing forest shares (Figure 2c). For example, when increasing the FLR option of natural succession of abandoned land, the agroforestry share comprised 36% to 41% of the remaining landscape (Figure 2e). In contrast, when forest share was constrained to less than 30% of the landscape, agroforestry comprised only 29% to 38% of the

remaining landscape (Figure 2c), whereas agroforestry dominated the remaining landscape when forest share was 30% or larger.

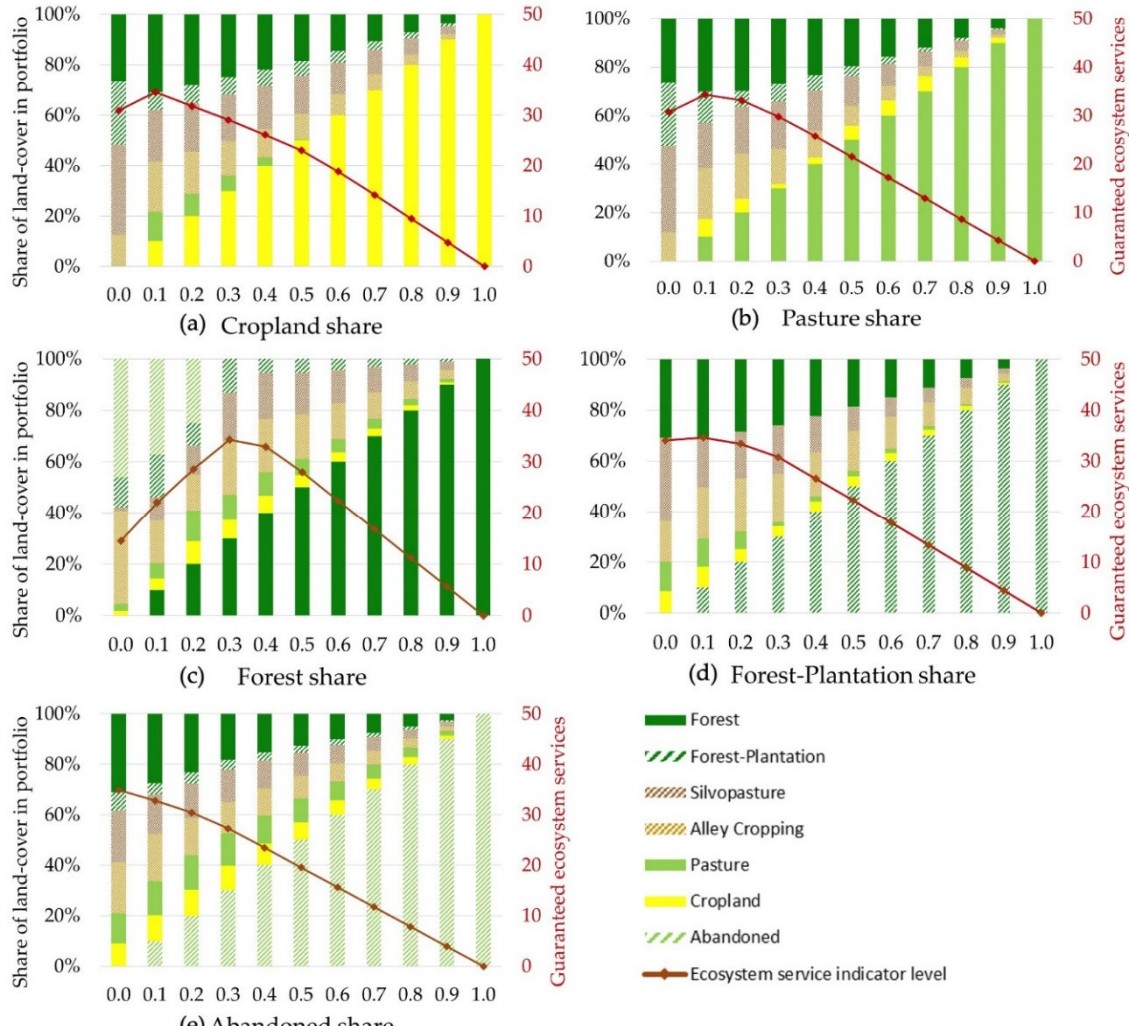

**Figure 2.** Impact of progressively expanding shares of (**a**) cropland, (**b**) pasture, (**c**) natural forest, (**d**) forest-plantation or (**e**) natural succession of abandoned land on land-cover composition (bars, left *y*-axis) and guaranteed level of ecosystem service indicators (min $\{p_{iu}\}$, see Equation (8), red line, right *y*-axis). The gradual increase of land-covers in the model is reflected by the steps (*x*-axis). Depicted land shares represent optimal landscape compositions according to the multi-objective optimization approach for a moderate level of uncertainty (*fu* = 2).

Hence, landscape composition influences the optimal share of agroforestry under a moderate level of uncertainty, but due to its high perceived performance agroforestry, shares of at least 34% were always selected to balance multiple ecosystem services at the landscape scale irrespective of the landscape context.

*3.3. Impact of Promoting Agroforestry*

In this section, we test the effect of promoting agroforestry on the composition and diversification of the remaining landscape. We also explore how promoting agroforestry would influence ecosystem service provision of the entire landscape (Figure 3).

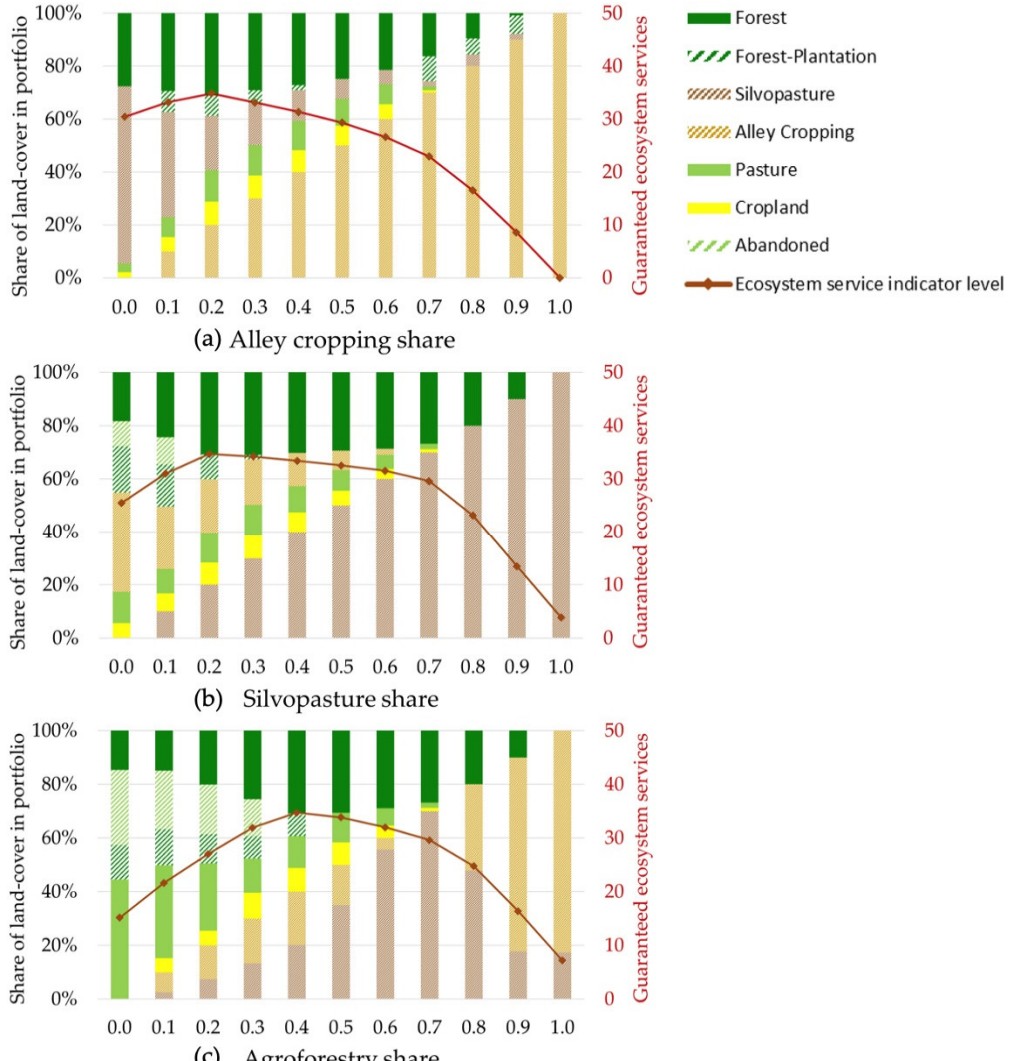

**Figure 3.** Impact of progressively expanding shares of (**a**) alley cropping, (**b**) silvopasture, and (**c**) both agroforestry systems combined on land-cover composition (bars, left *y*-axis) and guaranteed level of ecosystem service indicators (min {$p_{iu}$}, see Equation (8), red line, right *y*-axis). Landscape portfolios resulted from multi-objective optimization for a moderate level of uncertainty (*fu* = 2).

Interestingly, the forest share of the remaining landscape portfolio increased as agroforestry became more dominant in the landscape. When progressively increasing the share of alley cropping, the forest share increased until alley cropping comprised 70% of the landscape, at which point forest-plantation partially substituted forest (Figure 3a). Forest also dominated the remaining land-cover portfolio when the silvopasture share was above 40%, and replaced all other land-cover types when silvopasture comprised 80% of the landscape (Figure 3b). The development of the forest share was similar when the total area of agroforestry (alley cropping and silvopasture combined) progressively expanded (18% to 100% forest share of the remaining landscape (Figure 3c)).

We observed that increasing shares of agroforestry tended to homogenize the remaining landscape portfolio. Our results show that landscape diversity of the remaining optimized landscape decreased with increasing shares of agroforestry. For example, when silvopasture was restricted to 10% and lower, the landscape diversity was high (Shannon index: 1.62 to 1.68 (Table S4)) and decreased faster than for other land-cover types with increasing share of silvopasture. Similarly, when alley cropping and silvopasture together made up more than 30% of the landscape, the diversity of the remaining landscape declined (Shannon index: 0 to 1.52 (Table S4)). In contrast, when increasing agricultural

land-uses, forest or the two other FLR options, the diversification of the remaining portfolio remained relatively stable (Table S4).

We also found that the type of agroforestry affected the composition and level of diversification of the remaining portfolio: for very high shares of alley cropping (share > 80%), the model suggests diversifying the remaining landscape portfolio with forest-plantation, silvopasture and forest (Figure 3a). In contrast, in a silvopasture-dominated landscape, the model recommends a less diversified land-cover mix with forest making up the remaining land (Figure 3b).

Furthermore, our results suggest that silvopasture may be more suitable than alley cropping as a compromise solution. Ecosystem service provision tended to be higher when increasing the share of silvopasture in the portfolio compared to alley cropping. This is reflected by the higher guaranteed level of ecosystem services provided from silvopasture shares of 30% and larger (Figure 3a,b, red line, right *y*-axis).

Generally, agroforestry-dominated landscapes provided better solutions to balance multiple ecosystem services compared to landscapes dominated by other land-cover types. For example, when the model landscape was dominated by large shares of single agroforestry systems, the performance of the optimized landscape portfolios decreased more slowly with increasing share of agroforestry (compare Figures 2 and 3, red line, right *y*-axis). A landscape with a share of 70% alley cropping still provided multiple ecosystem services at a guaranteed level of 23%, while a landscape with 70% cropland could only guarantee a level of 14% (Figures 2a and 3a). Furthermore, we find that excluding both agroforestry types (Figure 3c, first bar on the left) would reduce the level of guaranteed ecosystem services provided (right *y*-axis: 15% ecosystem service level) to the same level of complete deforestation (Figure 2c, first bar on left).

Hence, landscapes with larges shares of agroforestry showed a tendency to conserve larger shares of natural forest while maintaining a high landscape performance, but tended to homogenize the remaining landscape in favor of tree-based land-cover types.

## 4. Discussion

### 4.1. The Role of Agroforestry in an Uncertain Multifunctional Landscape

In the face of global problems such as feeding a growing population while maintaining ecosystem functioning and biodiversity, allocating scarce land to various land-cover types has been a challenging task, which has led to controversial proposals such as giving half of our world's surface back to nature [69]. Our research approach allows decision-makers to explore the optimal mix of agroforestry and other FLR options in varying landscape contexts to meet a set of predefined objectives (10 ecosystem services in our case). We offer a decision support tool to explore the role of agroforestry and other FLR options for sustainable landscapes. It is particularly suitable in the common situation of scarce empiric data. Existing and hypothetical land-cover types can be considered while accounting for uncertainty of those land-covers in providing different ecosystem services.

Regarding our first research question, our results show that agroforestry was a particularly desirable FLR option to balance ecological and socio-economic ecosystem services at the landscape scale, based on current expert perception. In our survey, agroforestry was ranked higher than the alternative FLR options of forest-plantation and natural succession of abandoned land. Despite this clear expert judgement, agroforestry did not dominate the optimized land-cover portfolio under a moderate level of uncertainty; however, silvopasture and alley cropping did constitute a 41% share. Hence, the inclusion of both agroforestry systems in a FLR mix could lead to much higher guaranteed levels for all ecosystem service indicators compared to the optimized portfolio without agroforestry and the actual landscape portfolio (Figure 1). Despite half of the ecosystems service indicators reflecting socio-economic objectives, the optimized landscape only contained small shares of pasture and cropland. This reflects experts' positive judgment of the socio-economic potential of agroforestry, which replaced other agricultural land-uses in the optimized land-cover composition.

Furthermore, to obtain high ecological performance at the landscape scale, our model suggests that including agroforestry in the land-use mosaic might avoid the need to leave large areas as unmanaged abandoned land. These findings are in line with other studies that demonstrate the advantage of agroforestry in enhancing landscape multifunctionality [5,70].

Furthermore, the share of agroforestry was affected by the degree of uncertainty assumed. To avoid underperformance of ecosystem service indicators, our model suggested an increase in compositional diversity with increasing level of uncertainty. Increasing uncertainty increases unfavorable deviations of ecosystem services provided in worst case scenarios. Therefore, the model selects more land-cover types to buffer against poor performance of individual objectives. This effect can be explained by the averaging or portfolio effect and is in line with findings from land allocation studies in Ecuador [20,35].

We found that the current landscape composition of the study region performed poorest in terms of securing economic stability and food security, suggesting that these two ecosystem services require most attention in landscape planning. Integrating agroforestry in a landscape mosaic may contribute to objectives of food security and stable economic returns as shown by our results and those of other studies [4,5,71].

Our sensitivity analysis provides insights into the land-sharing vs. land-sparing debate. Land-sharing and land-sparing may both contribute to a multifunctional landscape. However, regarding our second research question, we found that agroforestry was always included in the landscape irrespective of the landscape context to meet multiple objectives under uncertainty (Figure 2). Thus, the model suggested that mixing the strategies land-sharing and land-sparing would lead to optimal results, holding that the landscapes consists of a high degree of compositional diversification of different land-cover types with large shares of agroforestry. Combining both strategies is in line with Meli et al. [12], who recommend that FLR needs to be implemented in both land-sharing and sparing. Runting et al. [72] also found that neither strict land-sharing nor land-sparing are desirable, when aiming for a multifunctional landscape. However, Paul and Knoke [73] point out that landscape diversification on separate pieces of land can still increase provision of multiple ecosystems services, without the establishment barriers associated with agroforestry systems, such as increased management complexity. Paul et al. [31] have also shown that a mixture of trees and crops on separate plots might be economically favorable for very risk-averse farmers, for instance those who depend heavily on income from their farm.

However, agroforestry may be promoted to diversify reforestation options in Panama, due to the reforestation project "Alianza por el Millón". Regarding the third research question, alley cropping and silvopasture showed slightly different impacts on landscape allocation and performance. Our results showed a higher suitability of silvopasture as a compromise solution compared to alley cropping. This is in line with Gosling et al. [65], who found that farmers in eastern Panama rated silvopasture higher than alley cropping across a range of socio-economic and ecological criteria.

Providing high levels of multiple ecosystem service indicators under uncertainty requires a high degree of compositional diversification within the landscape and/or at the plot level (i.e., agroforestry). For example, in a pasture-dominated landscape, the land-cover compositions would include forest, agroforestry, cropland and forest-plantation to balance ecological and socio-economic objectives under uncertainty. In contrast, when silvopasture was the dominant land-cover type, the remaining landscape would consist of forest, alley cropping, and agricultural land-uses or consist of natural forest only for very large shares of silvopasture. Thus, increasing silvopasture tended to homogenize the remaining landscape composition in favor of forest. Similarly, landscapes with a share of 30% to 80% of alley cropping supported forest shares in the remaining portfolio of above 40% (Figure 3). Other studies also reported that land-sharing may support forest conservation. For example, Angelsen and Kaimowitz [74] state that in contrast to highly intensified agricultural systems, agroforestry may reduce pressure on forests by increasing ecological and socio-economic benefits. By increasing long-term productivity, agroforestry may counter land degradation, thereby reducing land abandonment and the need to convert forest into productive agricultural land [74].

Regardless which land-use strategy is followed, landscape scale heterogeneity is important to support the provision of multiple ecosystem services [75]. As illustrated in previous studies [28,76], we show that a multifunctional landscape is best supported by heterogeneity in our example of a diverse landscape mosaic. Homogenous landscapes dominated by one or two agroforestry systems may have detrimental effects for multifunctionality, as reflected by our sensitivity analysis and other studies [75]. Regulations and incentives should be in place to encourage a mix of FLR options (including different agroforestry types) to support the development of a diverse landscape [77]. Furthermore, promoted agroforestry types should align with the needs of local farmers to facilitate adoption [78].

While the goal of our study was to find optimal landscape compositions that enhance the achievement of multiple ecosystem services at a tropical forest frontier, it remains unclear how enhancing the landscape performance would impact deforestation in the long run. Although, our results showed a trend that forest cover could even be increased in a multifunctional landscape including agroforestry, market dynamics might result in further agricultural expansion, if those competing land-uses prove to be more profitable than natural forest [20]. Mitigating tropical deforestation is a major global challenge. Our approach can contribute to understanding the consequences of considering multiple ecosystem services and uncertainty for landscape planning and deforestation.

## 4.2. Combining Expert Opinion and Multi-Objective Optimization

We emphasize that our input data for the multi-objective optimization is based on surveys with experts in their respective fields. This means that the data will be affected by personal perception and should be carefully interpreted. Although certain types of agroforestry (e.g., living fences and scattered trees in pasture) are common in our study area and Panama, the alley cropping and silvopasture systems considered in this study are not widespread, which may limit experience-based expertise of some survey participants. However, when compared to empiric findings at other sites in the tropics, the judgments of experts concerning the provision of ecosystem services for land covers seem plausible. Forests and tree-based systems were ranked highest for ecological indicators, in line with findings by Potvin et al. [79] for Panama and databases used by the IPCC [80]. In terms of food security, the two agroforestry systems received the top rankings. Alley cropping ranked highest, followed closely by conventional cropland and pasture, which aligns with findings by Reed et al. [4]. In the literature, combining trees and agricultural systems on the same piece of land may enhance ecosystem services [70] and increase resilience against extreme weather compared to conventional agricultural systems [58,71,81].

As tree products provide additional farm revenue, it seems plausible that agroforestry systems and forest-plantations were ranked highest for long-term profit by survey participants. This is in line with bio-economic modelling in the study area [31]. For long-term profit, forest was ranked similarly to cropland and pasture, which may be due to the perception of forest as a land-cover having no ongoing management costs but having the potential to sell firewood. In terms of liquidity, pasture (followed by cropland) was ranked higher than the agroforestry types. This seems plausible because cattle can be sold at any point in time [82,83] and trees represent a long-term investment [84]. Experts ranked agroforestry types highest for economic stability (even before forest-plantation). This may reflect that the agroforestry types are polyculture systems, whereas forest-plantations in our study represented a monoculture. Furthermore, agricultural revenues can be generated during the year through an agroforestry system, whereas exotic timber is best harvested after ca. 25 years from an economic perspective [50]. Ratings for scenic beauty could reflect the experts' personal preferences towards forest and agroforestry systems.

However, experts may have overestimated the advantages of agroforestry (particularly alley cropping) and underestimated its disadvantages. For example, Clough et al. [85] found for Indonesia that rubber production was lower in the agroforestry system compared to the monoculture system and generated considerably less income. A review conducted by Reed et al. [4] on the contribution of trees in the tropics worldwide found that studies reported both positive and negative effects of the trees on

food yield and overall livelihood. Despite both agroforestry types being ranked highly during our surveys, neither alley cropping nor silvopasture (according to our definition) are prevalent in our study region. The high ranks assigned to agroforestry systems might be due to the fact that agroforestry has become quite popular in science and politics and could reflect desirable thinking (e.g., [1–3]). However, it may also indicate that agroforestry systems are highly valued among the stakeholder groups in our study, but farm level constraints may prevent adoption, such as implementation costs [12], loss of agricultural production, investment costs in inputs and labor [70], and perceived investment risks [86]. Therefore, including calculated socio-economic indicators that reflect those potential farm level constraints may yield a different landscape composition with lower agroforestry share. However, the aim of this study was not to derive an optimal landscape composition from a farmer's perspective, but from the perspective of society.

While quantitative empiric data are valuable, they can be costly and time consuming to obtain. Using expert knowledge as input data for optimization has been shown to lead to similar results as measured or calculated data [49]. For example, in Uhde et al.'s [49] study, the share of a near-natural secondary forest was similar for the landscape portfolio based on expert opinion and the related variability (34% forest share) and the portfolio based on measured or calculated data and the corresponding uncertainties (29% forest share). Therefore, our model and results can provide a sound basis for further discussions with stakeholders regarding land-use planning for multifunctional landscapes.

However, our method to quantify expert knowledge using AHP also has its challenges. To illustrate, the number of land-cover and ecosystem service indicators which can be investigated is limited, because an increasing number of alternatives rapidly increases the number of pairwise comparisons which can make the survey time-consuming and tedious [43]. Including more land-covers in the study design may have resulted in a different landscape composition. However, we were prevented from including more alternatives because the length of the AHP survey would have become prohibitive.

As an alternative to using AHP, monetary values may be used to express ecosystem services provided across different land-covers [87–89]. By using monetary valuation, non-market goods may be excluded [90]. Alternatively, a combination of field measurements, model results, economic evaluation, survey data and calculations may be applied [91]. However, these approaches were not appropriate for our study because of data gaps, as we specifically wanted to test an approach under the common situation of data scarcity that allows land-use types that are not yet widespread or common in a given area to be included in the analysis.

We selected a robust multi-objective optimization approach to derive theoretical optimal landscape portfolios, because it supported our research aim and allowed to integrate uncertainty. Incorporating uncertainty in the modelling process is important when there is a lack of certainty about the demand and provision of ecosystem services [92]. We actively incorporated (dis)agreement in expert opinion about the provision of ecosystem services across different land-cover types into the optimization procedure. Such disagreements can be difficult to quantify in a group discussion, which are often used for ecosystem service valuation and prioritization studies (e.g., [93,94]). But it has a direct impact on the derived land-cover composition and may be an important piece of information for robust land-use planning. Disagreement in expert opinion is reflected by variation in land-cover scores. Higher disagreements are represented by higher standard deviations. This makes the respective land-cover less attractive for a risk-averse decision-maker. We focused on minimizing underperformance in worst-case scenarios by incorporating the negative (unfavorable) deviation from the expected mean, as opposed to accounting for both favorable and unfavorable deviations [35,49].

Another advantage of our optimization model is that it can be used for multiple and single objective optimization. In this study, single objective optimization allowed us to investigate the optimized landscape performance separately for each ecosystem service indicator. Furthermore, single objective optimization can be used to analyze which individual indicators are influencing the landscape portfolio.

In our optimization model, we assumed equal demand for all ecosystem services to avoid subjectivity [95] and therefore weighted the indicators equally. Weighing of indicators can reflect that some indicators may be valued higher than others. Although it was not the aim of our study, our approach allows for reflecting preferences of stakeholders through putting weights on specific indicators (see [28,65]). In the absence of determined weights for each ecosystem service, Gourevitch et al. [96] used an efficiency frontier for two objectives to display the range of preferences from valuing one objective over the other to 100%. However, since we considered more than two objectives and lack information of stakeholders' long-term preferences and constraints, we opted for equal weights [28]. Nevertheless, the current landscape composition of the study area diverged strongly from the optimized landscape, which indicates that current land-use decisions may not be driven by providing all 10 studied ecosystem services simultaneously at their best possible levels, but perhaps by a subset of our studied indicators. For example, Gosling et al. [65] showed that farmers' land-use decisions might be driven by more immediate objectives, such as meeting household needs and maintaining liquidity. However, predicting the current land-use allocation was not the intention of this study. We aimed to find a multifunctional landscape that meets the objectives of all stakeholder groups simultaneously.

Our results should not be interpreted as generally true for all of Panama. However, our findings regarding the positive perception of agroforestry and interrelations of agroforestry, other FLR options, agricultural land-uses and natural forest can be important for landscapes beyond our study area in eastern Panama. Even though quantitative empiric data are certainly favorable as a foundation for land-use planning, integrating expert knowledge into landscape planning can give important insights into general relationships to guide further research.

### 4.3. Opportunities for Future Research

Potential drawbacks of agroforestry (e.g., high investment costs and delayed financial returns) may lead to farmers rejecting sustainable land-use concepts based around agroforestry. Therefore, future studies may include greater consideration of farmers' objectives, perceptions and local knowledge. Bringing together scientific and experience-based knowledge can help find landscape compositions that reconcile competing demands of the public and private landowners [97].

It has been suggested that landscape planning for multifunctional landscapes today and in the future should account for landscape composition and configuration [12,98]. As a first step, our model investigated landscape composition. However, future landscape configuration should be considered for a holistic land-use plan and for investigating the impact of fragmentation on landscape multifunctionality. Fragmentation effects and impacts of adjacent land-covers on ecosystem services and biodiversity might be substantial [99]. To consider landscape configuration, spatially explicit models have been used [100] to map ecosystem services provided by different land-cover types based on expert knowledge [46] or monetary estimation [88]. Spatially explicit modelling may be crucial, when mapping potential costs and benefits of forest landscape restoration options that are spatially heterogeneous [96].

However, focusing on landscape composition instead of spatial configuration demands less computational power. This is preferable as long as spatial configuration is not expected to affect the results [101]. For instance, Duarte et al. [98] and Verhagen et al. [102] both emphasize the effects of compositional diversity on ecosystem service provision. In their reviews they found that only few services, such as nutrient retention, pollination or landscape aesthetics were found to be affected by configurational aspects. Hence, for most of the services investigated here, a linear relationship with area proportions may be assumed, given a relatively large landscape. Yet, this may be questioned for aspects such as biodiversity, water regulation and scenic beauty. Future studies could incorporate such aspects, for example through coupling the optimization approach with spatial simulation approaches or transferring the problem into more complex mixed-integer programming (see review [32]). While we used simplified indicators to reflect studied objectives, this approach would also allow for a better

representation of biodiversity-related objectives. In a next step, biophysical characteristics may be considered to determine where exactly land-sharing or sparing is appropriate [12]. This could support site-specific land-use planning. Biophysical aspects such as soil condition and economic aspects, such as investments costs, could be incorporated in our model, e.g., through including additional constraints which the optimized landscape composition must not violate.

Another important aspect for future research is time dynamics. While our model approach was static to optimize land allocation for the highest and most stable level of ecosystem services, future research may involve dynamic modeling. This could involve integrating time dynamics into evaluation of land-cover types [42] and modelling deforestation scenarios for tropical forests [28]. Temporal aspects might include seasonal fluctuations of ecosystem service provision [42], development effects (e.g., abandoned land turns into forest, altering its contribution to climate regulation; crop growth alters water regulation), climatic change and degradation effects. By integrating uncertainty into our optimization, we account for some volatility in delivery of ecosystem services and anticipate worst case scenarios.

Furthermore, future studies may test different shapes of uncertainty space to enhance precision and reduce data demand. Our model considered uncertainty boxes. Alternative uncertainty space shapes include conic spaces [64,103].

## 5. Conclusions

Combining the analytic hierarchy process and robust optimization, we were able to investigate stylized landscape compositions that theoretically provide multiple ecosystem services under uncertainty at the forest frontier based on expert perception. Our approach may contribute to a better understanding of interrelations between land-covers (prevalent and potential) and uncertain provision of different ecosystem services encountered in the common situation of scarce data. Using underperformance of ecosystem service provision as a measure, the model suggests establishing a mix of different land-covers with large shares of agroforestry in this example tropical landscape. For our study region, agroforestry was perceived by experts from different backgrounds and stakeholder groups as a key strategy to provide multiple ecosystem services, though it is not currently present in the study area. However, to improve landscape management, agroforestry systems (i.e., alley cropping and silvopasture) may best enhance multifunctional landscapes as a complement within a land-cover mosaic irrespective of the landscape context, leaving room for both land-sharing and land-sparing strategies [104]. This includes FLR options in an agriculture-dominated landscape, which may increase socio-economic indicators in particular, such as economic stability, food security and long-term profit, according to our results. Promoting agroforestry, as might be the case with Panama's reforestation initiative, may benefit forest and productive tree-based land-uses. However, measures against landscape homogenization may be considered to guarantee multiple ecosystem services.

We suggest that our approach, as a preliminary study, may help decision-makers to systematically analyze which mix of agroforestry and other FLR options may be best-suited under different conditions to foster a multifunctional landscape. Our approach, which is parsimonious in its data needs, may inform feasibility studies to derive insight into desirable forest landscape restoration concepts and landscape compositions. This helps to set priorities for further field-based research to investigate where exactly to put what kind of restoration, in terms of biophysical and economic considerations [11], and set priorities for funding specific options [86].

**Supplementary Materials:** The following are available online at http://www.mdpi.com/2071-1050/12/15/6077/s1, Method S1: Interviews to determine current landscape composition, Method S2: AHP survey, Figure S1: Example of AHP survey illustrating scale transformation, Table S1: Comparison of aggregated mean scores and standard deviations between all survey participants and participants with consistency ratio (CR) ≤ 20%, Table S2: Number of survey responses by ecosystem service indicator and survey group. Derived from AHP survey, Figure S2: Comparison of results from online and face-to-face surveys, Figure S3: Simplified example of uncertainty scenarios (u) for a robust multi-objective optimisation including three land-covers, Table S3: Coefficient of variation for land-cover types and ecosystem service indicators. Derived from AHP survey and based on Table 3 (see main

text), Table S4: Shannon index values of the remaining landscape portfolios. When share of different land-covers (top row) are restricted between 0% and 90% of the landscape area (left column), Figure S4: Optimal landscape composition under increasing levels of uncertainty. Optimal share of land allocated to each land-cover type to provide multiple ecosystem services, based on expert knowledge, Figure S5: Individual ecosystem service indicator performance. Normalized indicator values achieved (piu) over different uncertainty scenarios per indicator for (a) optimized landscape portfolio including all seven land-cover types, (b) excluding both agroforestry systems at *fu* = 2 and (c) current landscape, Figure S6: Optimal land allocation and landscape performance for achieving each ecosystems service indicator individually.

**Author Contributions:** Conceptualization, C.P.; methodology, T.K.; formal analysis, E.R.; investigation, E.G. and E.R.; writing—original draft preparation, E.R.; writing—review and editing, E.R., E.G., T.K. and C.P.; visualization, E.R.; supervision, T.K. and C.P.; project administration, C.P.; funding acquisition, C.P. All authors have read and agreed to the published version of the manuscript.

**Funding:** This research was funded by the German Research Foundation (DFG), grant number PA3162/1.

**Acknowledgments:** We thank the German Research Foundation (DFG) for funding the present research on the Potential of Agroforestry in Panama (PA3162/1). We particularly thank Andrés Gerique whose comments on the survey design were of great help. We are also grateful to all the participants who took part in the surveys and thank Alyna Reyes, Peter Glatzle and Rodrigo Duarte for helping to conduct the surveys. Finally, we thank Dominik Holzer for assistance with figures, Karen Grosskreutz and Alena Anderson Chilian for language editing, and the anonymous reviewers for their valuable suggestions.

**Conflicts of Interest:** The authors declare no conflict of interest and the funders had no role in the design of the study; in the collection, analyses, or interpretation of data; in the writing of the manuscript, or in the decision to publish the results.

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
