# Peer review of "How Much Agroforestry Is Needed to Achieve Multifunctional Landscapes at the Forest Frontier?—Coupling Expert Opinion with Robust Goal Programming"

_sustainability, doi:10.3390/su12156077_

Round 1

Reviewer 1 Report

  1. There is more about context needed here in the introduction – there is needed something about Panama and their deforestation and forests.

  1. Methods like mean-variance optimization or robust portfolio optimization need more explanations about these methods in the introduction section.

  1. There is needed something more about AHP in the introduction.

These three points are important to increase clarity of manuscript and its values and also interests to readers.

Study area: what is study are in ha?

In M&M the calculation of AHP should be briefly explained – this is the core of these measure, so maybe it will be better to see this in the text than in supplementary data.

What kind of exactly environment for programming was used?

I do not know if ‘liquidity’ is a good phrase used in proper context in this manuscript.

In equation 9, I do not see “l”.

Results: I think that clarity of text will increase if authors kindly provide a table with indicators used in this text and their descriptions (eg. for “fu”).

The idea of uncertainty should be better explained in the manuscript.

I think that meaning of this work for global deforestation problem (and also for Panama) should be more discussed in the discussion section.

What can be done if you want to know the farmers perspective? (in future research?)

Reviewer 2 Report

The focus of this paper on agroforestry as a component of Forest Landscape Restoration is welcome, as this issue has been unaddressed in the literature. It is very worthwhile to explore the role of alternatives to pure agriculture and plantation forestry (although formal alley cropping and silo-pastoral systems are uncommon; what is more common are more traditional systems like home gardens and trees in pastures). The methods seem sound, and the analysis seems well-executed. The conclusions are interesting, but I feel they are of limited practical utility because the paper is built on a weak empirical foundation.

The paper begins by talking a lot about integrating ecological and socio-economic objectives, But in fact, it is only about ecosystem services (a highly utilitarian, production-oriented view). The authors basically ignore biodiversity except to admit that they incorrectly subsume that under ecosystem services (lines 154-156).

Looking for an optimized land use mix suggests that land use is planned, rather than an emergent property of individual decisions, power dynamics, and history. Most landscape planning exercises in the study region are simply paper exercises with little on-the-ground application. This is not to say that it is not useful to examine the environmental service implications of different land use mixes, but it's purpose and likely usefulness should be more clearly acknowledged.

Surveying experts is a valid approach. But the authors note that agroforestry "is not common in the study region" (line 168), so how much do experts really know about it? I think this apples to both the ecosystem service an socio-economic benefits. I am not sure how much real experience-based expertise is represented in the responses.

Reviewer 3 Report

The general comment is that there are other relevant literature which the manuscript may benefit from. For mosaic pattern of landuse, there are Satoyama Index which evaluate on agroforestry types of landuse. The analysis of landscape indicators might give insights on the analysis of the site contexts and landuse planning. Satoyama Index is used in the following resesrch. Typology of cities based on city biodiversity index: exploring biodiversity potentials and possible collaborations among Japanese cities https://www.mdpi.com/2071-1050/7/10/14371 As regards the landscape values, the authors can consider and mention the qualitative ecosystem services which are not easily analyzed by quantitative methods. A literature which is discussing the landscape values is listed in the following. An explorative analysis of landscape value perceptions of naturally dead and cut wood: a case study of visitors to Kaisho Forest, Aichi, Japan https://www.tandfonline.com/doi/full/10.1080/13416979.2020.1773619 Figure 2 and 3 show interesting results. However, not only the sum of individual ecosystem service levels but the levels of categorized ecosystem services can be shown in the figures. For example, ecological and socio-economic categories ecosystem service levels can be separately indicated and discussed.

Reviewer 4 Report

REVIEWER STATEMENT - MS "sustainability-865055"

 The research topic is interesting and fits to the scope of the journal. The research objective is presented and responded adequately. Data and methods are carefully and thoroughly described. Results are manifold, and presented through relevant figures and tables, and commented pretty well in related texts. Discussion and conclusions are sound.

Some small comments:

  1. I would suggest to reorganize the introduction a bit. It would be, in my opinion, better to start from general agroforestry topics, definitions and challenges and then go to the study area peculiarities. THis can then be followed by methodological part, as has been done in the current version.
  2. In results (Figure 1) abandoned land use gets relatively high share, if agroforestry is not among among land-use options. Please try to comment this resul bit more.
  3. In the discussion, authors refer to to spatially more explicit approaches, which is a natural way to continue. Already in this study they could have (perhaps) considered that the target area may limit the total area of certain land-use options. For example, it may not be possible to implement agroforestry throughout the area due to e.g. site fertility or other constraints.

Round 2

Reviewer 3 Report

The manuscript has improved substantially reflecting the comments by the reviewers and it is almost ready for publication. This is rather a minor comment but the authors may wish to refer to "social– ecological production landscape (SELPs)" in explaining Table 1 and in Discussions when considering the trade-offs and synergies of different services. The ecological and socio-economic elements are split, yet the overall intention here is to analyze the overall interactions. There are relevant discussions in Journal of Forest Research or International Journal of Environmental Research and Public Health (by MDPI) and other journals including synergies and trade-offs of various ecosystem services.